# Repeated Manure Application for Eleven Years Stimulates Enzymatic Activities and Improves Soil Attributes in a Typic Hapludalf

Paulo A. A. Ferreira [1] , Mariana V. Coronas [1] , Max K. L. Dantas [2], André Somavilla [2] , Gustavo Brunetto [2],
Carlos A. Ceretta [2], Sandro J. Giacomini [2], Paulo I. Gubiani [2] , Gustavo Boitt [3] , Claudio R. F. S. Soares [4],
Glaciela Kaschuk [5] , Samya U. Bordallo [6] and Cledimar R. Lourenzi [6,*]

[1] Academic Coordination, Federal University of Santa Maria, Cachoeira do Sul 96506-322, RS, Brazil;
ferreira.aap@gmail.com (P.A.A.F.); mariana.coronas@ufsm.br (M.V.C.)
[2] Department of Soil Science, Federal University of Santa Maria, Santa Maria 97105-900, RS, Brazil;
maxdantas22@gmail.com (M.K.L.D.); somavillaa@gmail.com (A.S.); brunetto.gustavo@gmail.com (G.B.);
carlosceretta@gmail.com (C.A.C.); sjgiacomini@ufsm.br (S.J.G.); paulogubiani@gmail.com (P.I.G.)
[3] UWA School of Agriculture and Environment, The University of Western Australia, Perth 6009, Australia;
gustavo.boitt@uwa.edu.au
[4] Department of Microbiology, Immunology and Parasitology, Federal University of Santa Catarina,
Florianopolis 88040-900, SC, Brazil; crfsoares@gmail.com
[5] Post-Graduate of Soil Science, Federal University of Paraná, Curitiba 80035-050, PR, Brazil;
glaciela.kaschuk@ufpr.br
[6] Department of Rural Engineering, Federal University of Santa Catarina, Florianopolis 88034-000, SC, Brazil;
samyauchoa2000@gmail.com
* Correspondence: lourenzi.c.r@ufsc.br

**Abstract:** Animal manure may be a valuable resource for the development of agricultural sustainability. We proposed to verify the feasibility of applications of three types of animal manures to improve soil attributes and to sustain crop yields under intensive cropping and no-tillage systems. The field experiment was established in 2004 on Typic Hapludalf soil with pig slurry (PS), cattle slurry (CS), pig deep-litter (PL), mineral fertilizer (MF) and a non-fertilized treatment. From 2004 to 2015, were grown black oat, maize, forage turnip, black beans, and wheat. Soil samples were taken after winter 2014 and summer 2015, and submitted to chemical, physical, microbiological and biochemical analyses. Animal manures increased soil pH, but MF caused acidification of soil. The PL and CS applications reduced soil density, and increased total pore volume and hydraulic conductivity. Animal manures increased soil P fractions, total organic carbon, total nitrogen, stimulated soil respiration, and had higher activities of glucosidase and acid phosphatase. Wheat had its biggest dry matter and grain yields with MF, but maize grain yields with CS were higher than MF. All indicators pointed that application of animal manure converges to an interesting strategy to recycle nutrients at farmyard level and to contribute to global sustainability.

**Keywords:** acid phosphatase; glucosidase; microbial phosphorus stock; maize; wheat; grain yield

## 1. Introduction

The recent increase in world trading and increased world economic growth has led to tremendous increases in the global consumption of meat and animal products, particularly in developing countries [1]. In that context, traditional meat producer countries have intensified their production systems to meet the demands for meat and many other animal products to the international market. For example, in 2019, Brazil had a livestock of approximately 215 million cattle and 41 million pigs, accounting, respectively, for 14 and 5% of the total cattle and swine livestock of the world [2]. Considering that cattle produce approximately 5.5 kg of dry mass manure per animal per day, with moisture contents varying from 13 to 75% [3], and an adult swine may produce 8.5% of its body weight

in manure [4]. In 2019, Brazil may have produced more than one Megaton of dry mass cattle manure and, assuming an average of 25 kg per pig, another two Megatons of pig slurry per day. Slurry is a waste product that has to be removed from breeding sheds to improve animal welfare and health. When accumulated in one place, it may cause severe environmental pollution, particularly when it moves into water bodies, where eutrophication causes environmental disequilibrium. It is usually deposited in landfilling or spread on agricultural fields, pastures, and tree plantations to decrease environmental pollution, sometimes treated as waste material rather than a valuable resource.

Humankind has used animal slurry as manure and watered their crops for at least 8000 years, since 6000 BC [5], but with the advent of the Green Revolution in the 1960's, farmers used alternative mineral fertilizers to accomplish the goal of improving soil nutrient status [6]. Recently, several meta-analyses have emphasized the benefits of animal manure application on crop yields and chemical, physical and biological soil attributes [7–13]. Animal manure of many kinds are expected to improve soil quality and crop yields on organic production systems, where mineral fertilizers are avoided, but also in intensive agricultural production systems [14–18].

A limitation may be the mystification that the application of animal manure has too many disadvantages, making farmers resign their traditional ways of fertilizing the soil with slurry. For instance, the addition of animal manure alone could represent a risk of lowering crop yields in relation to mineral fertilizers because with organic fertilization the nutrients are released slowly during the completion of the immobilization–mineralization processes [1,12,19]. The nutrient concentrations in organic fertilizer are relatively low but they perform important functions compared to chemical fertilizer. Organic fertilizers and their proper management may reduce the use of chemical fertilizers. Hence, the small farmers can save the cost of production. On the other hand, nutrients from organic sources are released slowly in the soil environment which makes them available for a longer period of time and helps to maintain soil nutrient status. However, animal manure is a source of C, which is a limited resource for soil microorganisms [9,20,21]. The addition of manure should stimulate soil microbial biomass and activity, and increase activities of the enzymes related to the biogeochemical cycles of nutrients [9,11,12,21,22]. While in the short term, the addition of manure could stimulate the degradation of soil organic matter and increase the depletion of soil organic C due to higher soil respiration [12,21], repeated applications should accumulate more soil organic matter in the long term, which would have a crucial role on the improvement of chemical and physical soil attributes [10,12,16,23]. Therefore, even intensive agriculture, based on the full package of the Green Revolution, would be benefited by application of animal manure.

The use of animal manure as fertilizer for annual crops is not always a viable option in agricultural production areas because collection, preparation and distribution of manure can be expensive when transporting over long distances is necessary [3,15,24,25]. However, in cattle, swine and poultry producing regions, accumulated manure is a reality that has to be managed [24]. Several studies have shown that the excessive accumulation of manure in a single location can have serious environmental impacts, mainly due to the displacement of nutrients and eutrophication of water bodies [15,24,25]. However, measuring and understanding the benefits of incorporating organic manures even in conventional intensive systems may give subsidies for the development of public policies, which may change the scenario of discouragement and support more sustainable development. Understanding the dynamics of microbial activity in these systems is, therefore, desirable for designing successful management strategies aiming to optimize nutrient availability and improve plant productivity.

In this study, we proposed to analyze several indicators of soil quality, and to verify the feasibility of three types of animal manure (pig slurry, cattle slurry and pig-deep-litter) applied yearly during 11 years to improve chemical, physical and biochemical soil attributes and to sustain crop yields under intensive cropping and no-tillage systems.

## 2. Materials and Methods

### 2.1. Long-Term Field Experiment

The field trial was conducted from 2004 to 2015 in an area under no-tillage with 16 applications of animal manure at the experimental land of Soil Department of the Federal University of Santa Maria (UFSM), Rio Grande do Sul state, South of Brazil (29°42′50.97″ S, 53°42′25.10″ W). The climate of the region is humid subtropical (Cfa 2), with an annual average temperature of 19.3 °C. Average annual rainfall is 1561 mm and relative humidity of 82%. The mean rainfall, maximum and minimum air temperatures, and mean soil temperature in the study period are shown in Figure 1. The soil of the experimental area is classified as a Typic Hapludalf [26]. In the year of implementation, the soil presented the following physical-chemical characteristics in the 0–0.10 m layer: 108 g kg$^{-1}$ clay; 183 g kg$^{-1}$ silt; 709 g kg$^{-1}$ of sand; 22 g kg$^{-1}$ of organic matter; pH-H$_2$O 4.65; 23 mg kg$^{-1}$ of P and 32 mg kg$^{-1}$ of K (extracted by Mehlich-1); 0.30 cmol$_c$ dm$^{-3}$ Al; 0.65 cmol$_c$ dm$^{-3}$ Ca and 0.38 cmol$_c$ dm$^{-3}$ Mg (extracted by KCl 1 mol L$^{-1}$).

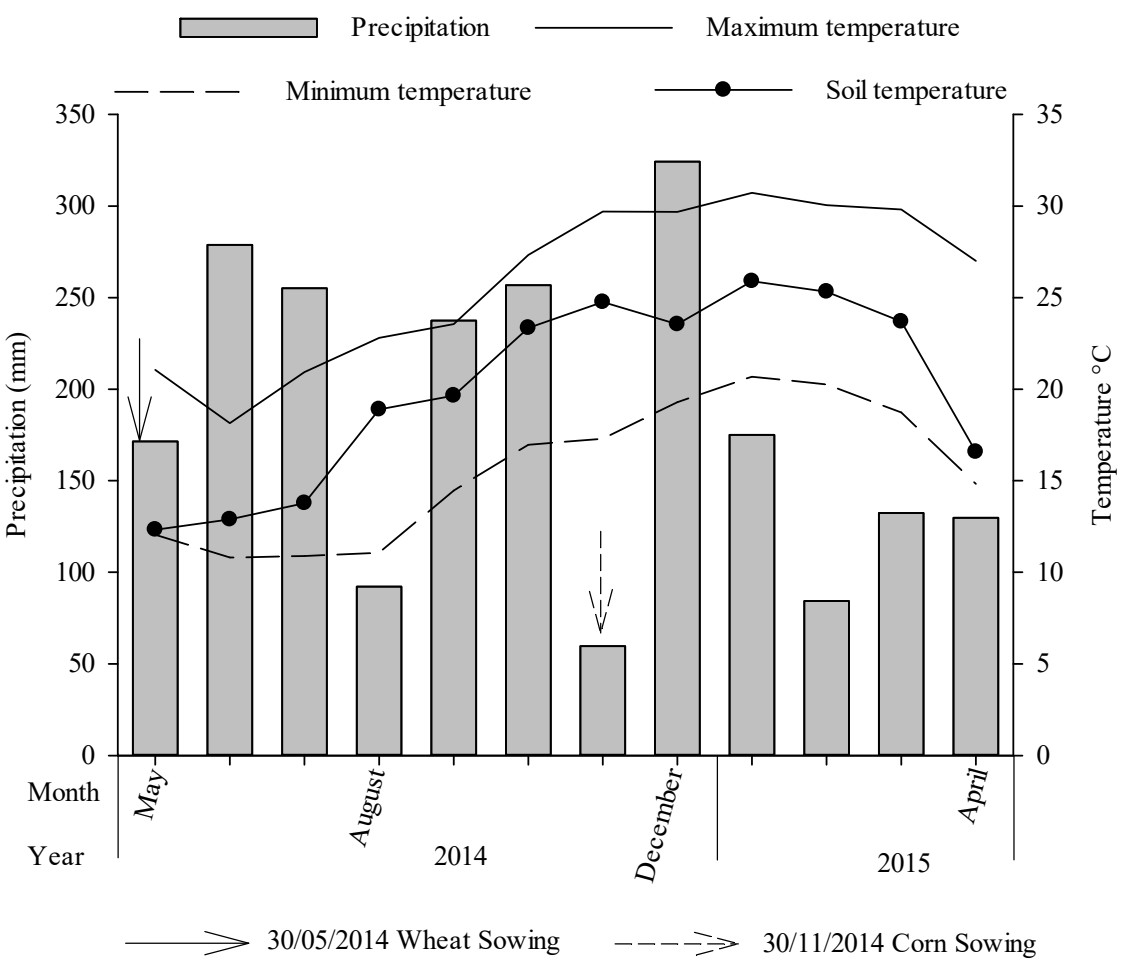

**Figure 1.** Average precipitation, maximum and minimum air temperature and average soil temperature, in the agricultural year of 2014/2015 in the area where the experiment was conducted.

The experimental design was a randomized block with four replications and plots with 25 m$^2$ (5 × 5 m). Five treatments were tested: pig slurry (PS), cattle slurry (CS), pig deep-litter (swine manure + rice husk) (PL), mineral fertilizer (urea + triple superphosphate + potassium chloride) (MF) and a control treatment (without nutrients input). The pig slurry and cattle slurry consisted of feces, urine, food scraps and water used to clean the facilities. The pig deep litter consisted of rice processing residues, feces, urine and food scraps. The applied rate of each organic residue was defined based on the N concentration

and its requirement by maize (*Zea mays* L.) and wheat (*Triticum aestivum*), according to the regional recommendation [27]. PS, CS, PL and MF were soil surface broadcasting on the crop residues (without incorporation). Urea, triple superphosphate and potassium chloride were used as sources of N, P and K in the NPK treatment, respectively. Before sowing, wheat received 60 kg P ha$^{-1}$ and 58 kg K ha$^{-1}$, while maize received 30 kg P ha$^{-1}$ and 58 kg K ha$^{-1}$. In this treatment, N fertilization was split in two applications: 50 kg N ha$^{-1}$ at sowing and 50 kg N ha$^{-1}$ 35 days after sowing, respectively. In wheat cultivation, 23 m$^3$ ha$^{-1}$ of PS, 103 m$^3$ ha$^{-1}$ of CS and 19 t ha$^{-1}$ of PL were applied and in corn crop 58 m$^3$ ha$^{-1}$ of PS, 102 m$^3$ ha$^{-1}$ of CS and 18 t ha$^{-1}$ of PL were applied.

The crops succession used between 2004 and March 2013 was: black oat (*Avena strigosa* Schreb.), maize (*Zea mays* L.), forage turnip (*Raphanus sativus* L.) and black beans (*Phaseolus vulgaris* L.), as shown in Figure 2.

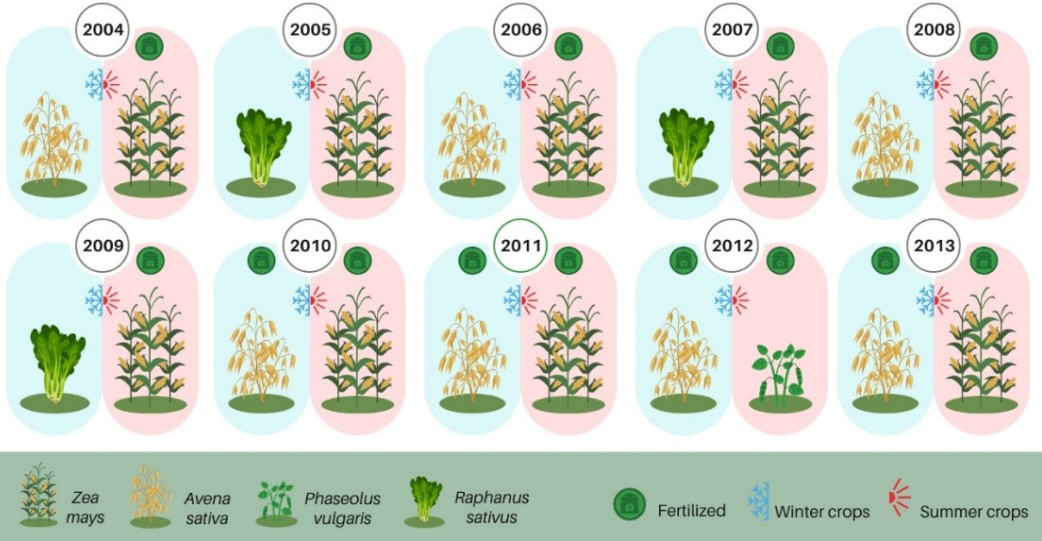

**Figure 2.** Schematic representation of the cropping systems with winter crops and of summer crops season in Southern Brazil.

In 2014, wheat was grown in the winter and maize in the summer. The wheat, cultivar TBIO Pioneiro was seeded at a density of 350 seeds m$^{-2}$, and the maize, cultivar DKB 240 PRO3 was seeded with 5 linear plants m$^{-1}$ spacing at 0.70 m between rows. The total amounts of dry matter, N, P and C added by the organic waste, mineral fertilizer and the crop residues are shown in Table 1.

**Table 1.** Total dry matter (TDM), total carbon (TC), total nitrogen (TN) and total phosphorus added by different organic waste, mineral fertilizer and crop residue in the period preceding this study, i.e., from 2004 to June 2014, totaling 14 applications of animal manure.

| Treatments [1] | Total Added by Fertilizers | | | | Total Added by Crop Residue | | | |
|---|---|---|---|---|---|---|---|---|
| | TDM | TC | TN | TP | TDM | TC | TN | TP |
| | t ha$^{-1}$ | | | | | | | |
| Control | - | - | - | - | 68 | 33.358 | 0.519 | 0.184 |
| MF | - | - | 1.120 | 0.576 | 106 | 54.252 | 0.874 | 0.348 |
| PS | 18.1 | 4.780 | 1.439 | 0.822 | 147 | 74.307 | 1.267 | 0.466 |
| CS | 67.7 | 15.258 | 1.942 | 0.765 | 123 | 63.415 | 1.120 | 0.413 |
| PL | 134.9 | 35.419 | 1.894 | 1.133 | 131 | 66.467 | 1.223 | 0.450 |

[1] Control, Mineral fertilizer (MF), Pig Slurry (PS), Cattle Slurry (CS) and Pig deep-litter (PL).

### 2.2. Soil and Plant Tissue Analyses and Grain Yield

In 2014, during the flowering season of wheat and maize, the soil was sampled in the 0–0.10 m topsoil layer, according to the regional recommendation [27] (with Soil Profile Sampler—50 mm Diameter) to quantification of biochemical and microbiological soil properties. Samples were conditioned in sterile plastic bags and kept under refrigeration (4 °C) prior to analysis. The microbial C (C-BIO), P (P-BIO) and N (N-BIO) were determined by the fumigation/extraction method [28,29]. The stocks of C (C-BIO), P (P-BIO) and N (N-BIO) were calculated according to the expression: Stock (C-BIO), (P-BIO) and (N-BIO) = (C-content; P-content; N-content) × (Ds × Zf), where: Stock (C-BIO), (P-BIO) and (N-BIO): the stock of carbon, phosphorus and nitrogen in the microbial biomass at a certain depth; Ds: the soil density at a depth of 0–10 cm; and Zf: is the thickness of the considered layer.

Basal respiration (BR) was determined by capturing $CO_2$ evolved from the soil in 0.5 mol $L^{-1}$ NaOH solution after 7 days of soil incubation. The metabolic quotient ($qCO_2$) was estimated by the relationship between BR and C-BIO, while the microbial quotient (qMicro) was estimated by the relationship between C-BIO and total organic-C, both based on [30].

To evaluate the activity of soil microbial enzymes, the hydrolysis of fluorescein diacetate (FDA) was determined by colorimetry [31]. The activities of acid phosphatase, β-glucosidase and arylsulfatase were performed by colorimetry with p-nitrophenol. Urease was determined by the release of $NH^{4+}$ after the incubation of the soil with a urea solution [32]. At the end of the maize cycle, in 2015, the soil was sampled to determine the chemical properties: pH, available P, K, Ca, Mg, organic phosphorus, organic C and organic N [33]. As well, undeformed samples were collected to determine the following soil physical properties: density, total pore volume, macroporosity, microporosity, geometric mean diameter of aggregates, hydraulic conductivity and soil air permeability [33].

Plant dry matter production was quantified in the wheat and maize full flowering period. For wheat, an area of 0.25 $m^2$ was sampled and for maize, five plants per plot were sampled. The plants were dried in an oven with forced air at 65 °C until constant weight, ground and stored. Grain yield was determined from a working area of 3.00 $m^2$ for wheat and 6.25 $m^2$ for maize and the grain moisture corrected to 13%.

### 2.3. Statistical Analysis

Differences in means of soil and plant variables were tested by one-way ANOVA. The normality of the residues, homogeneity of variances and independence of means (Anova assumptions) were considered during the data analysis. When significative the means were compared by Scott Knott test (post-hoc; $p \leq 0.05$). Statistical analyses were carried out in Sisvar 4.0 for one-way ANOVA and post-hoc tests [34].

## 3. Results

### 3.1. Chemical and Physical Properties of the Soil

Mineral fertilizer decreased soil pH and base saturation, increased Al availability whereas animal manures conserved a higher pH. The addition of CS increased the contents of Ca and Mg in the soil. The P-total was higher in the soil that received PL applications (Table 2). The P-Mehlich-1 content was also higher with the continued use of PL, exceeding 2.73 times the MF P- Mehlich-1 content (Table 2). However, there was only an increase in the content of P-organic with the use of CS, which was 62 and 24% higher in relation to the control and MF, respectively. On the other hand, there was no difference in the levels of P-organic with the PS and PL in relation to MF. Total organic carbon (TOC) and total nitrogen (TN) were higher for the PL and CS. The C/N ratio did not present statistical differences between the treatments. In the case of the C/Organic-P ratio, it was higher in the treatment with the addition of CS.

**Table 2.** Chemical and physical attributes of a Typic Hapludalf collected at the end of the maize cycle in 2015, in the 0–0.10 m layer, after 11 years and 16 applications of animal manure.

| Treatments [1] | Control [†] | MF | PS | CS | PL |
|---|---|---|---|---|---|
| | | Chemical attributes | | | |
| pH | 4.97 b | 4.30 c | 4.99 b | 5.10 b | 5.40 a |
| Ca—$cmol_c$ $dm^{-3}$ | 1.22 d | 1.13 d | 2.61 c | 3.45 b | 5.52 a |
| Mg—$cmol_c$ $dm^{-3}$ | 0.62 a | 0.85 a | 0.87 a | 1.12 a | 1.46 a |
| K—mg $dm^{-3}$ | 23.10 d | 60.10 c | 73.40 b | 103.4 a | 111.0 a |
| P—mg $dm^{-3}$ | 8.71 d | 41.37 c | 71.49 b | 75.94 b | 93.10 a |
| Al saturation—% | 44.18 b | 66.30 a | 27.40 c | 22.42 c | 9.55 d |
| V—% | 13.61 c | 9.16 c | 22.1 b | 27.91 b | 52.49 a |
| P—Total—mg $kg^{-1}$ | 711 d | 856 c | 887 b | 995 b | 1257 a |
| P—Org—mg $kg^{-1}$ | 4.52 d | 27.51 c | 45.17 b | 31.32 c | 104.60 a |
| COT—g $kg^{-1}$ | 7.80 c | 9.40 b | 9.70 b | 12.60 a | 12.30 a |
| NT—g $kg^{-1}$ | 0.70 c | 0.90 b | 1.00 b | 1.30 a | 1.20 a |
| COT/NT | 11.14 a | 10.44 a | 9.70 a | 9.96 a | 10.25 a |
| COT/P-org | 1725 a | 342 b | 215 c | 402 b | 118 d |
| | | Physical attributes | | | |
| DS—g $cm^{-3}$ | 1.58 a | 1.51 a | 1.51 a | 1.44 a | 1.30 b |
| VTP—$cm^3$ $cm^{-3}$ | 0.40 c | 0.45 b | 0.45 b | 0.46 b | 0.56 a |
| Ma—$cm^3$ $cm^{-3}$ | 0.08 b | 0.12 a | 0.11 a | 0.09 b | 0.13 a |
| Mi—$cm^3$ $cm^{-3}$ | 0.32 c | 0.33 c | 0.34 c | 0.38 b | 0.43 a |
| DMG—mm | 0.49 a | 0.65 a | 0.69 a | 0.78 a | 0.76 a |
| Ksat—mm $min^{-1}$ | 0.13 b | 0.68 a | 0.53 a | 0.75 a | 0.72 a |
| Kar—$\mu m^2$ | 1.27 b | 7.60 a | 6.94 a | 8.01 a | 14.48 a |

Means followed by the same lower case letter in the column do not differ by the Scott Knott test ($p \leq 0.05$). [1] pH in water; P extracted by Mehlich-1; P-organic extracts by NaOH-EDTA; TOC: total organic carbon; NT: total nitrogen; C/P-org: Carbon/P-organic ratio; Ds: soil density; VTP: total pore volume; Ma: macroporosity; Mi: microporosity; DMG: geometric mean diameter; Ksat: hydraulic conductivity; Kar: soil permeability to air; [†] Control, Mineral fertilizer (MF), Pig Slurry (PS), Cattle Slurry (CS) and Pig deep-litter (PL).

The soil density (Ds) was lower with the application of PL and presented the highest total volume of pores (TVP) and microporosity (Mi). Macroporosity (Ma) was also higher with CS and MF (Table 3). Geometric mean diameter (GMD) did not change, but hydraulic conductivity (Ksat) and soil air permeability (Kar) were lower in the control treatment.

**Table 3.** Carbon, nitrogen and phosphorus in microbial biomass (C-BIO, N-BIO, P-BIO), respiration of soil microbial biomass (C-$CO_2$), metabolic quotient (q$CO_2$) and microbial quotient (qMicro) layer of 0–0.10 m of an Typic Hapludalf in samples collected in wheat and maize blossoms in no-tillage area after 15 and 16 applications of the treatments, respectively, during eleven years.

| | | | Wheat | | | |
|---|---|---|---|---|---|---|
| Treatments [1] | C-BIO (mg $kg^{-1}$ solo) | N-BIO (mg $kg^{-1}$ solo) | P-BIO (mg $kg^{-1}$ solo) | C-$CO_2$ (mg C-$CO_2$ $kg^{-1}$ solo $h^{-1}$) | q$CO_2$ ($\mu$g C-$CO_2$ $mg^{-1}$ C-BIO $h^{-1}$) | qMicro (%) |
| Control | 104.79 a | 26.45 b | 3.38 b | 0.25 b | 2.38 b | 1.34 a |
| MF | 121.22 a | 25.43 b | 5.96 a | 0.31 b | 2.59 b | 1.29 a |
| PS | 116.26 a | 31.49 a | 3.60 b | 0.31 b | 2.60 b | 1.20 a |
| CS | 126.40 a | 37.28 a | 5.09 a | 0.48 a | 3.79 a | 1.00 b |
| PL | 113.09 a | 34.77 a | 5.34 a | 0.49 a | 4.32 a | 0.93 b |
| | | | Maize | | | |
| | C-BIO (mg $kg^{-1}$ solo) | N-BIO (mg $kg^{-1}$ solo) | P-BIO (mg $kg^{-1}$ solo) | C-$CO_2$ (mg C-$CO_2$ $kg^{-1}$ solo $h^{-1}$) | q$CO_2$ ($\mu$g C-$CO_2$ $mg^{-1}$ C-BIO $h^{-1}$) | qMicro (%) |
| Control | 123.10 b | 16.97 c | 4.25 c | 0.25 b | 2.12 c | 1.63 a |
| MF | 120.84 b | 19.08 c | 10.96 a | 0.57 a | 4.72 b | 1.32 b |
| PS | 120.96 b | 16.04 c | 9.62 a | 0.42 b | 3.42 b | 1.35 b |
| CS | 158.88 a | 32.86 a | 12.09 a | 0.65 a | 4.12 b | 1.36 b |
| PL | 128.95 b | 23.71 b | 8.35 b | 0.74 a | 5.83 a | 1.01 c |

Means followed by the same lower case letter in the column do not differ by the Scott Knott test ($p \leq 0.05$); [1] Control, Mineral fertilizer (MF), Pig Slurry (PS), Cattle Slurry (CS) and Pig deep-litter (PL).



### 3.2. Microbiological Attributes of the Soil

The C-BIO levels were not altered by the treatments in wheat cycle. However, C-BIO was 30% higher using CS in maize cycle (Table 3). N-BIO was higher with the use of animal residues in wheat and in maize, except for PS. It is important to note that in both wheat and maize cycles there was no difference in N-BIO and C-BIO content with MF or control treatments. The lower contents of P-BIO were observed in the control treatment in both crops and PS (in wheat cycle) and PL (in maize cycle) were not differenced to the control. Microbial biomass respiration (C-$CO_2$) was higher with the PL and PS in wheat. However, in maize they did not differ with the MF. The long-term use of PL promoted the greatest metabolic quotient (q$CO_2$), although it did not differ from the soil that had been receiving additions of CS in wheat. The microbial quotient (qMicro) was bigger in the control, although it did not differ in the soil with the MF and PS in wheat.

The use of animal residues or MF did not differ in the capacity to increase the microbial C-stock in the soil during the growing season of wheat (Figure 3a). Nevertheless, in maize, the CS stood out (Figure 3b). The soil microbial N-stock was higher with the CS and PL in both crops (Figure 3c,d). However, soil microbial P-stock was not differenced with the use of animal residues and MF (Figure 3e,f).

In relation to soil enzymes activity, it was observed that FDA was higher for the CS, PS and MF in wheat (Figure 4a). The levels of soil enzyme activity (FDA) in maize cycle were similar between soils with PS, CS, PL and MF, and higher than those observed for the control (Figure 4c). The β-glucosidase activity in wheat was higher for the CS, reaching about 500% higher values than those observed for the control (Figure 4c). In maize, β-glucosidase activity was higher with CS and PL, and lower with PS (Figure 4d). The activity of β-glucosidase in the MF was similar to that of the control for the two crops (Figure 4c,d).

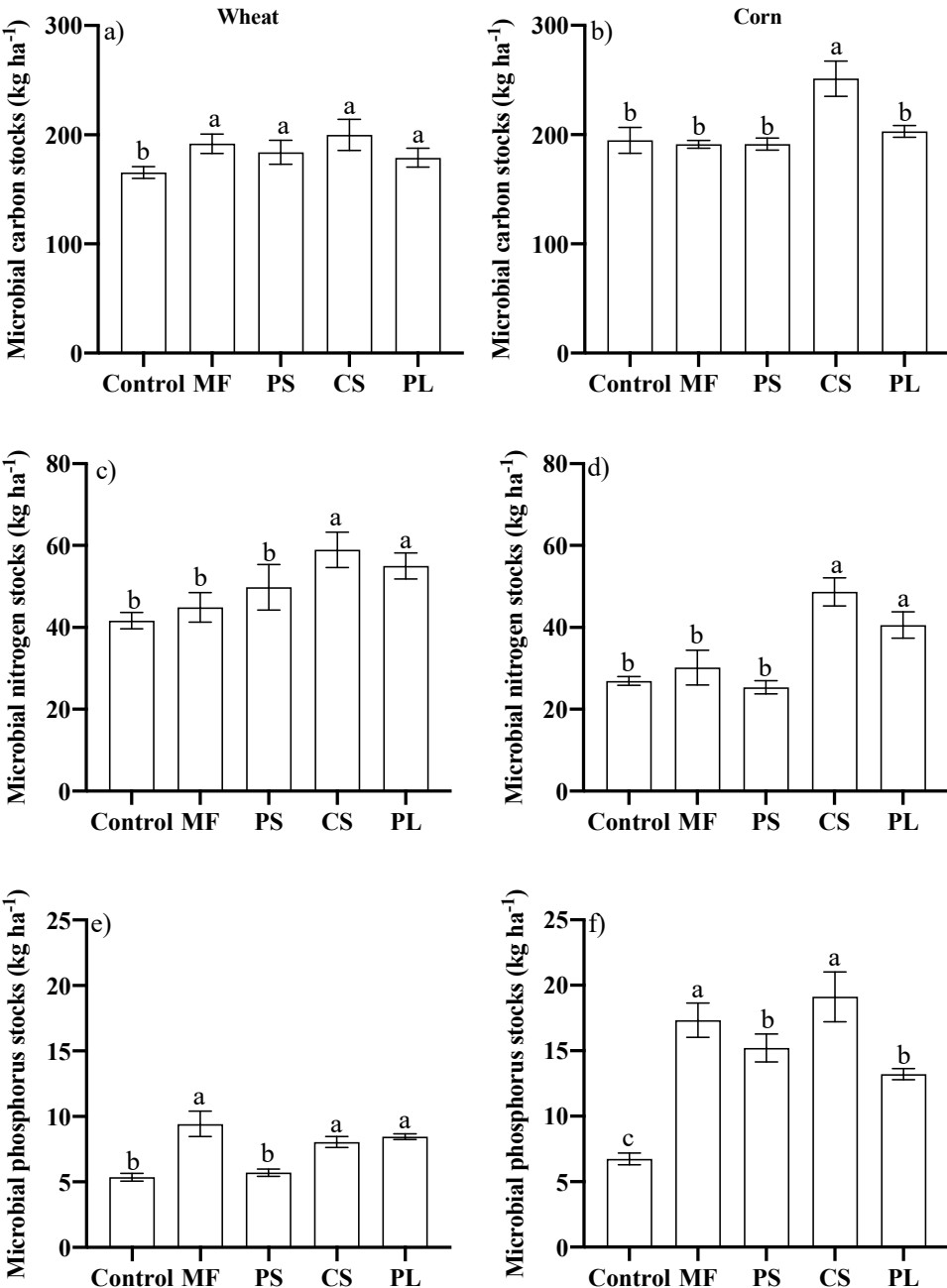

**Figure 3.** Stock of carbon (**a**,**b**), nitrogen (**c**,**d**) and phosphorus (**e**,**f**) in the microbial biomass in the 0–0.10 m layer of a Typic Hapludalf during the cultivation of wheat and corn, respectively, in no-tillage area after successive applications of different nutrient sources. Vertical bars with the same letter do not differ by the Scott Knott test ($p \leq 0.05$); Control, Mineral fertilizer (MF), Pig Slurry (PS), Cattle Slurry (CS) and Pig deep-litter (PL).

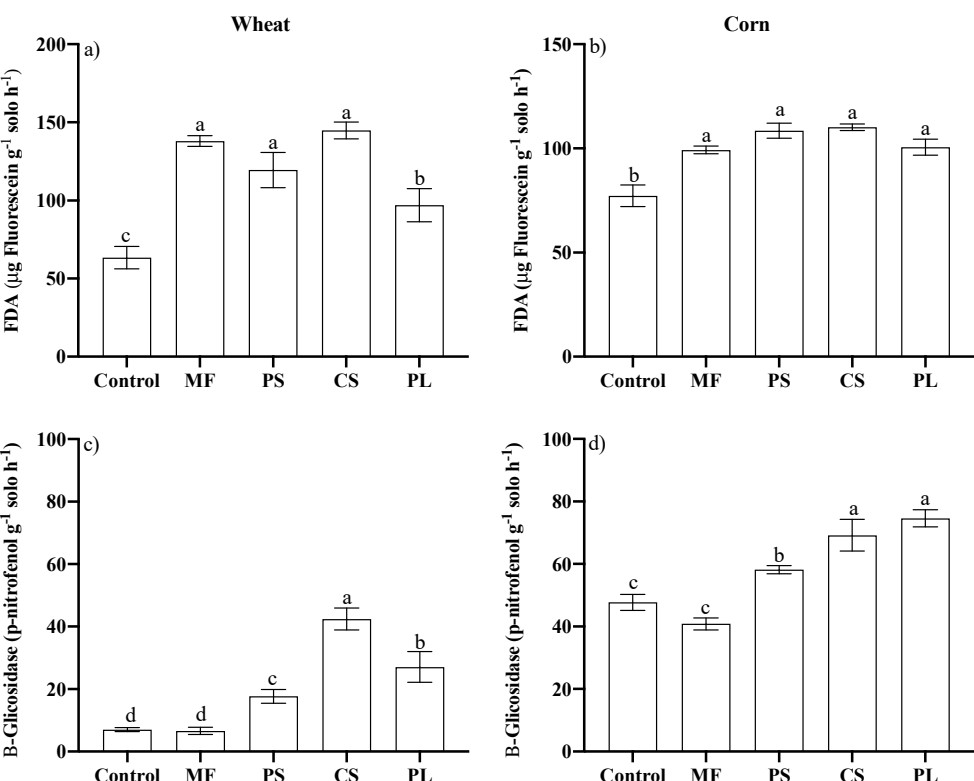

**Figure 4.** Activity of the FDA (**a**,**b**) and B-Glycosidase (**c**,**d**) enzymes in the 0–0.10 m layer of a Typic Hapludalf in samples collected in the wheat and corn blossoms, respectively, in no-tillage area after 15 and 16 treatments applications, respectively, for eleven years. Vertical bars with the same letter do not differ by the Scott Knott test ($p \leq 0.05$); Control, Mineral fertilizer (MF), Pig Slurry (PS), Cattle Slurry (CS) and Pig deep-litter (PL).

For both wheat and maize, the activity of urease was higher for MF and PL, although it did not differ from the CS (Figure 5a,b). The activity of acid phosphatase was higher in the soils that had received animal residues, especially CS, with higher activity in maize (Figure 5c,d). There was no variation in the acid phosphatase activity in the treatments MF and control, in both crops. The arylsulphatase activity was similar to that obtained for MF, PS and control in wheat (Figure 5e). On the other hand, for maize cycle the arylsulphatase activity was higher in control compared to the MF and PS (Figure 5f). The activity of arylsulfatase was higher in the PL, although it did not differ from the CS for wheat.

### 3.3. Dry Matter and Grain Yield of Wheat and Maize

Dry matter and wheat yield were higher with the MF. Among the organic fertilizers, PL and CS positively affected those parameters (Figure 6a,c). The MF increased wheat yields by 11, 16 and 62%, respectively, in relation to the PL, CS and the control, respectively (Figure 6c). In the case of maize, both dry matter and grain yield were higher with CS (Figure 6b,d), although in the case of grains it did not differ from the PS, with increases of 260 and 214% compared to the control, respectively, whereas PL provided similar results to MF (Figure 6d).

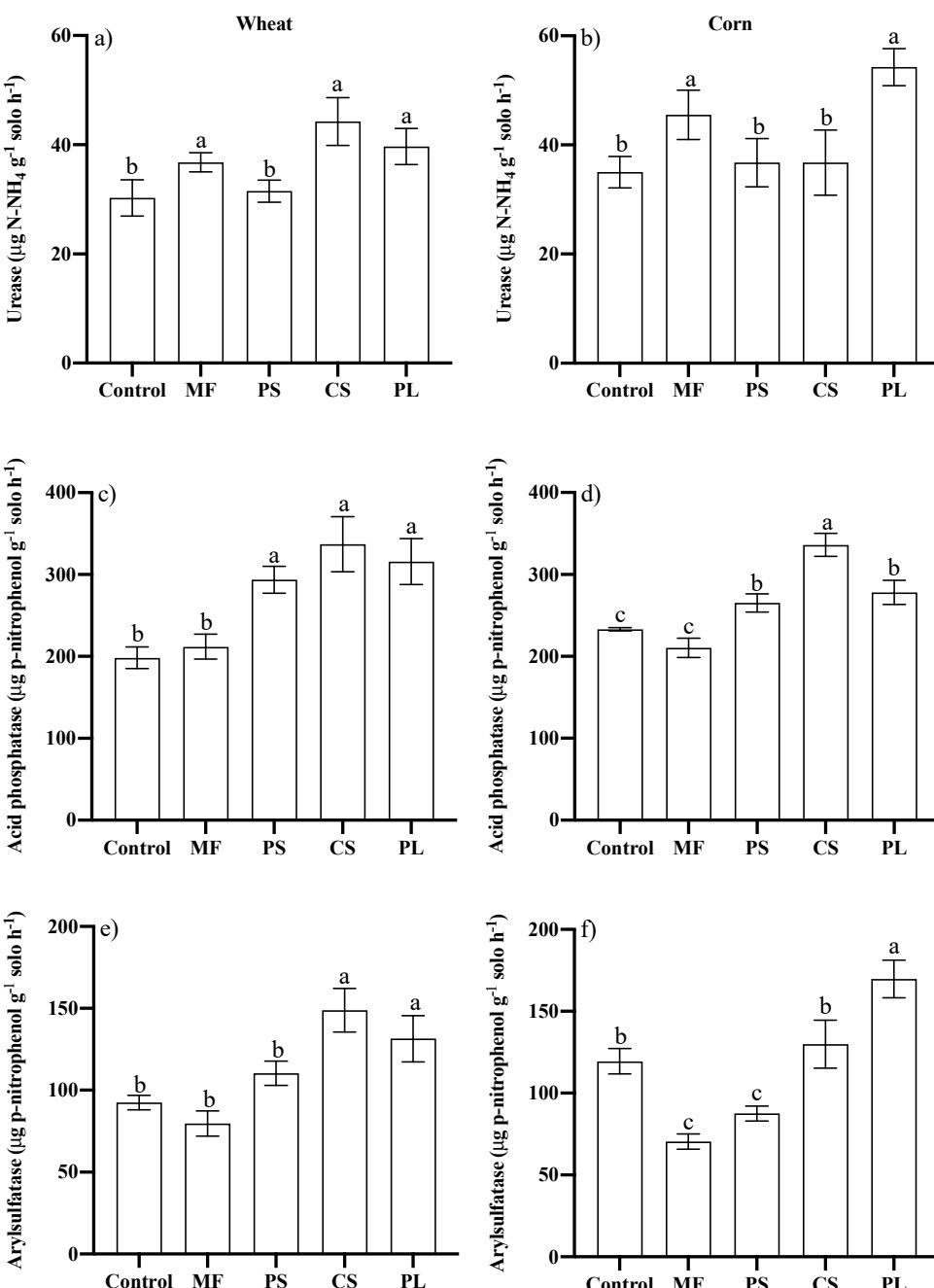

**Figure 5.** Activity of the enzymes Urease (**a**,**b**), Acid Phosphatase (**c**,**d**) and Arylsulfatase (**e**,**f**) in the 0–0.10 m layer of a Typic Hapludalf during the cultivation of wheat and corn, respectively, in no-tillage area after successive applications of different nutrient sources. Vertical bars with the same letter do not differ by the Scott Knott test ($p \leq 0.05$); Control, Mineral fertilizer (MF), Pig Slurry (PS), Cattle Slurry (CS) and Pig deep-litter (PL).

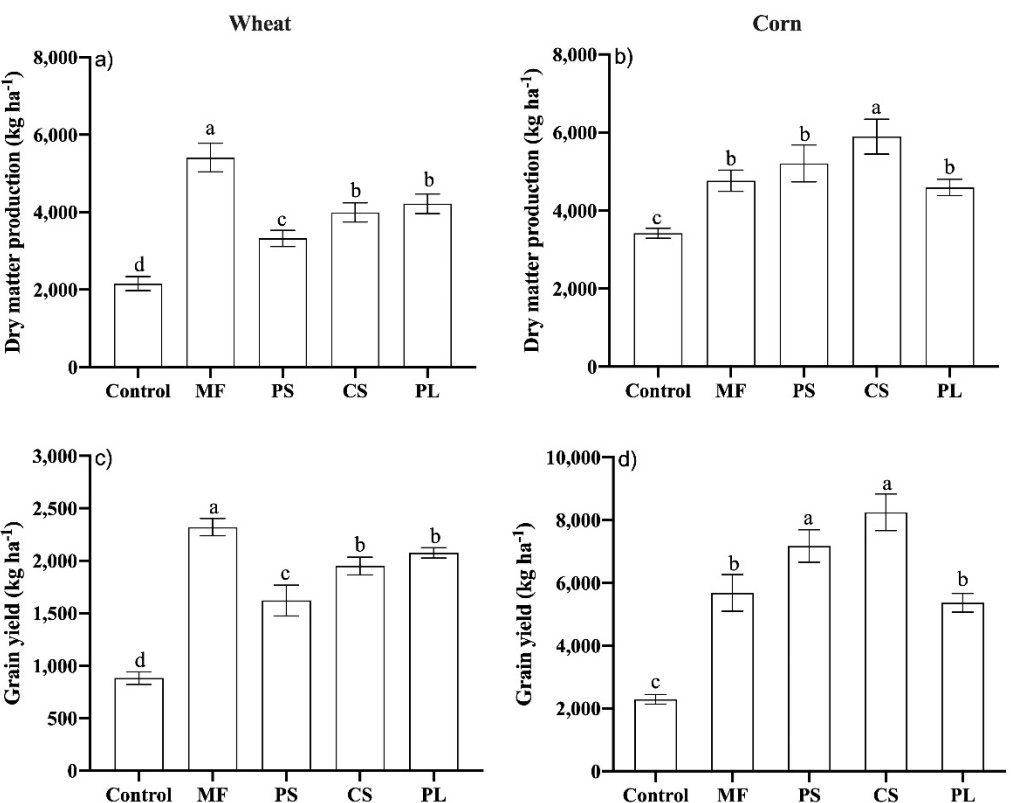

**Figure 6.** Dry matter production (**a**,**b**) and grain yield (**c**,**d**) of wheat and corn crops in no-tillage area after successive applications of different nutrient sources. Vertical bars with the same letter do not differ by the Scott Knott test ($p \leq 0.05$); Control, Mineral fertilizer (MF), Pig Slurry (PS), Cattle Slurry (CS) and Pig deep-litter (PL).

## 4. Discussion

In this manuscript, we show that animal manure could be part of intensive agriculture due to its positive effects on chemical, physical and microbiological soil attributes, essential for the maintenance of crop yields and soil quality.

First of all, there were very interesting effects of organic waste on the soil chemical attributes. The treatments with organic waste increased soil pH whereas the treatment with mineral fertilizer caused acidification of soil (Table 2), corroborating data obtained elsewhere [6,10,12]. Lower pH probably facilitated the solubilization of Al and decreased saturation of bases (Table 2), making the soil with MF less fertile than the soil with organic waste. Soil acidification is one of the main constraints of crop growth as it influences the availability of nutrients to plants, affects the functioning of roots [35], and of soil microbial biomass, important for establishing healthy biogeochemical cycles [6].

Both mineral and organic fertilizer improved the nutrient contents of P and K in the soil, but plots receiving organic waste were more fertile than plots with MF (Table 2). The experiment started with soil containing 23 mg kg$^{-1}$ of P-Mehlich-1 in 2004, and after 11 years repeating the treatments, the values of available P-mehlich-1 varied from 8.7 in non-fertilized plots, to 41.4 in mineral fertilizer-plots and to 93.1 mg kg$^{-1}$ of P in PL manure fertilized plots (Table 2). The improvement of P contents of soil is positive for crop growth, particularly in tropical soils where P is one of the limiting nutrients of crop yields [35,36]. Except for the non-fertilized, all treatments have presented very high availability of P, that is, >40 mg kg$^{-1}$, according to the Commission of Chemistry and Soil Fertility of RS/SC [27], but they were approaching the environmental risk thresholds [37–39]. The P added, in addition to meeting the demand of crops, partly remains on the soil and is considered the 'P legacy' [36]. The environmental risks of P legacy are due to losses through surface runoff (in particulate and dissolved forms) and subsurface movement [36], which could cause

eutrophication of water bodies. However, empirical data in similar areas of this experiment showed that P has dislocated only up to 30 cm through the soil profile [40]. In that situation, major environmental risks should occur with surface water runoff [41], and for that, one should consider that additional measures of soil conservation in the field are mandatory.

The application of CS and PL increased COT and NT in relation to PS, MF and control treatments (Table 2). The NT is a reservoir of N, which may become available for soil biota and plants. The COT is a reservoir of C for heterotrophic soil microbial biomass and also contributes to the complexation of undesirable elements, retention of nutrients and moisture, and soil structuring [8,12]. In fact, TOC may contribute to the binding of soil particles into aggregates, which rearrange spaces creating pores in the soil. The increases in TOC are often correlated with decreases in soil bulk density [8,12,23]. In this study, increases in COT (i.e., in PL treatment) were accompanied by increases in macro and microporosity and decreases in soil density (Table 2). The changes caused by application of PL on physical soil attributes are highly desirable for crop systems since less dense soils (and with more pores) result in higher water infiltration and aeration, with higher microbial and plant growth [8,12,19,23].

We estimated the soil microbial biomass by measuring the C, the N and the P (C-BIO, N-BIO e P-BIO) retained inside soil microbial cells [28,29] because their values may change quickly due to changes in the environment, so that, higher values indicated better soils for microbial and plant growth [42–44]. In this study, the application of CS increased C-BIO during maize growth and the application of CS and PL increased the N-BIO (Table 3), which indicates that the application of animal manure is improving soil conditions for microbial growth [9,19,45]. However, the application of CS and PL also increased the emissions of $CO_2$ per mass of microbial biomass, resulting in higher $qCO_2$ (Table 3). Commonly, literature relates higher rates $qCO_2$ with a more stressful soil environment because microorganisms are releasing more $CO_2$ than its baseline for growth maintenance [30,46]. However, in this study, the $qCO_2$ may be indicating that soil microorganisms are investing in more expensive metabolic processes, probably related to the mineralization of soil organic matter [12,19]. In that case, soils fertilized with organic waste are increasing their biological activity in relation to non-fertilized or mineral-fertilized plots.

Whereas the techniques of soil microbial biomass and soil respiration (Table 3) measure the abundance of living microorganisms in the moment of sampling [28,29], the measurements of activities of soil enzymes, which, are of biological origin but are not inside a cell anymore (Figures 4 and 5) makes an assessment of activity of past microbial communities, and how soil microorganisms have been working over a long time [7]. In this study, the activities of FDA hydrolysis, which is mediated by several oxidoreductase enzymes, were higher in both mineral and organic waste fertilized plots in both crops (Figure 4). It is possible that the higher crop growth (Figure 6), obtained in both mineral and organic waste fertilized plots in relation to non-fertilized plots, stimulated microbial activity because of higher development of roots and more root exudation [47]. On the other hand, the measurements detected more activities of B-glucosidase and acid phosphatase in the plots receiving any kind of animal manure in both crops (Figures 4 and 5).

Glucosidase and phosphatases are directly involved in the mineralization of soil organic matter, attacking C-compounds and organic P-compounds, respectively [11,22]. Since soil chemical analyses revealed that labile P was satisfactorily available in all fertilized plots (Table 2), it is possible that soil microorganisms were not aiming at P, but probably C. In another word, soil microorganisms may have adjusted their metabolism in order to acquire the most limiting nutrients and preserve their cell nutrient ratios [20,21]. Therefore, in this study, the addition of residues rich in organic-P compounds is possibly activating the recycling of soil organic matter, resulting in more $qCO_2$ (Table 3) but also, producing more mineralized nutrients (Table 2). As a matter of fact, several changes caused by the application of organic waste improved the biogeochemical cycle of P (e.g., total P, available-P, P-BIO, phosphatase activity). Considering that P is a limited resource [35,36],

using animal manure, where this resource is available, would be an interesting strategy to optimize the use of P at farm-level.

The benefits of organic waste to soil structure should converge to agricultural sustainability, meaning that crop yields are sustained over the years. Crop yields are determined by the accomplishment of several processes that depend on physical and biochemical factors. In this study, non-fertilized plots failed to have satisfactory dry matter and grain yields (Figure 6) as they were lower than official average crop yields in that year and region, i.e., 1330 kg ha$^{-1}$ of wheat and, 6300 kg ha$^{-1}$ of maize [48]. Wheat had its biggest dry matter and grain yields (almost 2500 kg ha$^{-1}$) when fertilized with MF and 2000 kg ha$^{-1}$ when fertilized with PL (Figure 6c). Fields relying on organic waste depend on soil microbial activity to acquire N, i.e., through mineralization of soil organic matter [6,49]. Microbial activity is regulated, among others, by temperature, soil humidity, pH and redox potential [50].

The lower yields of wheat under organic manure compared with mineral fertilizer may have been caused by low soil microbial activity due to lower temperatures (Figure 1). Indeed, soil microbiological analyses may support this hypothesis. First, the C-CO$_2$ emissions of the treatments receiving manure were lower in the wheat than in maize (Table 3), implying that soil microorganisms were less active [42,46]. Second, the microbial nitrogen stocks in the wheat were bigger than in the maize (Figure 3), implying that there was more immobilization of N and less mineralization in the soil microbial biomass during the wheat cropping. On the other hand, the yields of maize fertilized with CS were over 8000 kg ha$^{-1}$, which was much higher than the ones fertilized with MF (6000 kg ha$^{-1}$) (Figure 6). As temperatures and soil humidity were adequate for microbial growth, N was probably released in pace with the crop demands.

It should be considered that this study is reporting a complete shift from mineral fertilizer to organic waste soil fertilization, which revealed significant improvements in soil quality, but also limitations of winter temperatures for N mineralization. For future research development, one may consider a mixed cropping system including both mineral fertilizer and animal manure [1,6]. Partial doses of mineral fertilizers (for example, for N supply) may be included during period of lower soil microbial activity if the interest is to warrant crop yields [1,6]. Nevertheless, application of organic manures should increase C for soil microbial biomass, which retains and releases nutrients during the processes of mineralization of soil organic matter and microbial growth (immobilization), and represents a potential source of N and P bioavailable to plants because it is considered a labile fraction of soil organic matter [51]. The N-BIO stocks in the treatments with CS and PS and the P-BIO stocks in the treatments with MF and CS show that the soil microbial biomass represents an important pool for the soil stock of N and P. With this, some of the N and P can be released after microorganisms' death and become available to plants. The fraction of the microbial biomass is considered the living fraction of the organic matter due to its composition, and represents an important reservoir of C, N and P in tropical agricultural systems, containing around 1 to 5% C, 2 to 5% N and 2 to 20% P of the total element stock [52]. The beneficial effects of organic waste on microbiological soil indicators should contribute to agricultural sustainability due to increases in nutrient recycling in the soil and at farm-level.

## 5. Conclusions

The use of organic fertilizer for 11 years showed that it can contribute a lot to the quality of the soil, because it was able to provide increases of up to 22% in carbon contents and this was reflected in the enzymatic activity, which was increased by 52%, in addition to of the impact on N and P stocks. These variables were more positively impacted by the use of swine overlay (SO) in relation to mineral fertilizer. This increase in carbon content over these 11 years also contributed to a decrease in soil density and better hydraulic conductivity and permeability. On the other hand, the gains in crop grain productivity, which reached 20% in maize, showed that organic residues are more competitive with

mineral fertilization for crops in the summer period, due to the greater biological activity in the soil.

**Author Contributions:** Conceptualization, P.A.A.F. and M.K.L.D.; methodology, M.K.L.D., A.S., G.B. (Gustavo Boitt) and P.I.G.; validation, S.J.G., P.I.G., M.V.C. and C.R.F.S.S.; formal analysis, M.K.L.D. and A.S.; investigation, M.V.C., G.K., S.U.B. and C.R.L.; resources, G.B. (Gustavo Brunetto) and C.A.C.; data curation, P.A.A.F., M.K.L.D., G.B. (Gustavo Boitt) and P.I.G.; writing—original draft preparation, P.A.A.F., M.K.L.D. and G.K.; writing—review and editing, G.K., S.U.B. and C.R.L.; supervision, P.A.A.F.; project administration, P.A.A.F.; funding acquisition, G.B. (Gustavo Brunetto), C.A.C. and P.A.A.F. All authors have read and agreed to the published version of the manuscript.

**Funding:** This research was funded by Foundation for Research Support of the State of Rio Grande do Sul (FAPERGS) (process number 1971-2551/13-2) and the Brazilian National Council for Scientific and Technological Development (CNPq), process number 400982/2016-1.

**Acknowledgments:** We are grateful to the Foundation for Research Support of the State of Rio Grande do Sul (FAPERGS) (process number 1971-2551/13-2) and the Brazilian National Council for Scientific and Technological Development (CNPq), process number 400982/2016-1, for scholarships and funding available for this study.

**Conflicts of Interest:** The authors declare that they have no conflicts of interest.

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
