# Peer review of "Repeated Manure Application for Eleven Years Stimulates Enzymatic Activities and Improves Soil Attributes in a Typic Hapludalf"

_agronomy, doi:10.3390/agronomy11122467_

Round 1

Reviewer 1 Report

Detailed experimental methods should be introduced.

  1. what was difference between pig slurry and pig deep-litter?

What size of pig deep-litter used in field?

  1. to the table 1,did organic waste add in treatment of MF? If no, why MF had TN and TP from organic waste?
  2. how much PK was used in MF?
  3. how were the pig slurry (PS), cattle slurry 113 (CS), pig deep-litter (PL), mineral fertilizer added in field? The deep in field?
  4. how to sample the soils from field? Why only soils from 0-10 cm was sampled

The references should be revised according to journal style.

Author Response

1 - what was difference between pig slurry and pig deep-litter?

ANSWER. The pig slurry is applied in liquid form and the pig deep-litter in solid form. Changes made according to your suggestion. Please see lines 122 to 126.

What size of pig deep-litter used in field?

ANSWER. We add in the text the amounts of each organic source applied in the cultivation of wheat and corn. Please see lines 133 to 135.

2 - to the table 1, did organic waste add in treatment of MF? If no, why MF had TN and TP from organic waste?

ANSWER. We changed the information in the table that caused confusion.

3 - how much PK was used in MF?

ANSWER. Changes made according to your suggestion. Please see lines 131 and 132.

4 - how were the pig slurry (PS), cattle slurry 113 (CS), pig deep-litter (PL), mineral fertilizer added in field? The deep in field?

ANSWER. This information appears in lines 128 and 129.

5 - how to sample the soils from field? Why only soils from 0-10 cm was sampled

ANSWER. The cultivation system adopted in the experimental area is the no-tillage system, which does not carry out soil disturbance. As the application of treatments is on the soil surface, the greatest biological activity and the improvement of physical and chemical properties occur with greater intensity in this layer. In addition, the Rio Grande do Sul corrective and fertilizer recommendation manual defines soil sampling in a no-tillage area as a depth of 0 - 0.10 m.

Reviewer 2 Report

It was with pleasure that I studied the manuscript received for review. I consider the subject matter discussed as current and necessary. Replacing (or supplementing) mineral NPK fertilization with organic fertilizers is the basis for managing in a sustainable (pro-environmental) system. Particularly important is the effect of different fertilization (the best variant) on the physical, chemical and biological properties of the soil. The soil is the "starting point" for the size and quality of crops. The authors used wheat and corn as test plants. This is the right decision as they are widely cultivated in many countries around the world. The fertilization of these cereals was differentiated as follows: Control, Mineral fertilizer, Pig Slurry, Cattle Slurry and Pig deep-litter. The obtained research results are global (scientific and practical).

The comprehensiveness of this research should be appreciated. The authors determined, inter alia, 13 parameters of soil chemical properties, 7 - psyhiical attributes and biological and enzymatic properties of soil. The results are presented in an uncomplicated, legible and clear manner. Result tables and figures are not "overloaded" with a large amount of numerical data, which in my opinion is an advantage. I also have no objections to the interpretation of the results obtained. It has been proven that the adopted variants of organic fertilization used for 11 years significantly contribute to the improvement of most of the assessed soil parameters (especially the increase in the content of organic C in soil). This, in turn, positively influences the increase (by more than 50%) of the soil enzymatic activity, as well as the reduction of soil density. An important hint for agricultural practice is the fact that organic fertilization has a better effect on the productivity of maize than on wheat.

I have one comment to the article. In the chapter "Methodology", I do not understand the issue of the sequence of crops (forecrops for wheat and maize) in the 11-year research cycle. Were the forecrops in an identical arrangement for wheat and maize? Could you consider a graphical representation in this chapter of an experiment diagram or describe the issue more closely?

Author Response

REVIEWER #2

It was with pleasure that I studied the manuscript received for review. I consider the subject matter discussed as current and necessary. Replacing (or supplementing) mineral NPK fertilization with organic fertilizers is the basis for managing in a sustainable (pro-environmental) system. Particularly important is the effect of different fertilization (the best variant) on the physical, chemical and biological properties of the soil. The soil is the "starting point" for the size and quality of crops. The authors used wheat and corn as test plants. This is the right decision as they are widely cultivated in many countries around the world. The fertilization of these cereals was differentiated as follows: Control, Mineral fertilizer, Pig Slurry, Cattle Slurry and Pig deep-litter. The obtained research results are global (scientific and practical).

The comprehensiveness of this research should be appreciated. The authors determined, inter alia, 13 parameters of soil chemical properties, 7 - psyhiical attributes and biological and enzymatic properties of soil. The results are presented in an uncomplicated, legible and clear manner. Result tables and figures are not "overloaded" with a large amount of numerical data, which in my opinion is an advantage. I also have no objections to the interpretation of the results obtained. It has been proven that the adopted variants of organic fertilization used for 11 years significantly contribute to the improvement of most of the assessed soil parameters (especially the increase in the content of organic C in soil). This, in turn, positively influences the increase (by more than 50%) of the soil enzymatic activity, as well as the reduction of soil density. An important hint for agricultural practice is the fact that organic fertilization has a better effect on the productivity of maize than on wheat.

I have one comment to the article. In the chapter "Methodology", I do not understand the issue of the sequence of crops (forecrops for wheat and maize) in the 11-year research cycle. Were the forecrops in an identical arrangement for wheat and maize? Could you consider a graphical representation in this chapter of an experiment diagram or describe the issue more closely?

ANSWER. To improve understanding of the crop sequence, we made a graphic presentation as can be seen on page 4.
